# OpenReview forum: "Inference-Time Intervention: Eliciting Truthful Answers from a Language Model"
_NeurIPS.cc/2023/Conference — NeurIPS 2023 spotlight_

### Official Review · Reviewer_Zn8f · 2023-06-15

**Soundness:** 3 good
**Presentation:** 2 fair
**Contribution:** 2 fair
**Rating:** 7
**Confidence:** 4

**Summary:**

## Summary
The authors have introduced a new way to ensure LM generations provide truthful answers. They modify activations using a set of learned directions across top-K attention heads. The new method, "Inference Time Intervention," entails identifying a few attention heads with high classifier accuracy in linear probing. During the inference procedure, the technique changes the activations in directions related to the truth. It is a form of activation editing.

This paper focuses on improving the accuracy of LLMs - specifically, addressing cases where the model knows the correct answer but produces an incorrect one. The authors use the difference between generation and probe accuracy to operationalize what it means for the network to "know." Saunders et al. 2021 have also emphasized this point in their previous works.

### Dataset & Evaluation
In the paper, the dataset used is called the Truthful QA dataset. However, some of the questions in the dataset may cause the model to provide incorrect answers. The dataset has two types of answers: "truthful" (meaning not false) and "informative" (meaning answering the question). The metric "truthful informative" measures the percentage of truthful and informative answers to evaluate the model's accuracy.

**Strengths:**

- The paper is well-motivated and addresses a significant problem in alignment. The technique presented is simple, non-invasive, and data-efficient.

**Weaknesses:**

- See Questions below.

**Questions:**

- How did you decide on using 40 samples to construct the probing dataset? The train-to-validation ratio was 4:1. Were 32 examples used for training the classifier model and 8 for validation? It seems too low to qualify for a probe. Or was the probe classifier trained on 40 x n examples where n is the size of the Truthful QA dataset? The latter approach seems more reasonable technically. However, this joint training for all samples ignores the difference in discovered heads for different Truthfulness types, e.g., logical falsehoods, conspiracies, common points of confusion, etc. The head representing logical falsehood might differ significantly from the head representing conspiracy.
- Based on the results, it seems the Mass Mean Shift method for identifying the truthfulness direction gives the best results. Is there any similarities between the directions identified for different layers and heads?
- I am worried about using CE and KL to measure the model's divergence from the original. It suggests that this method ensures accurate results without deviating too much from the initial distribution. However, observing a decrease in CE and KL values is insufficient. We must also evaluate how this approach impacts the model's performance on other tasks. In particular, we need to analyze this intervention's adverse effects on the model's performance on downstream tasks.
- My previous point is of more significant concern because the evaluation was done automatically using GPT-judge instead of by humans.
- I need help understanding Figure 2(b). Based on its construction, it functions similarly to PCA. However, I need clarification as to why the major axis of the ellipses is all oriented in the same direction. I had anticipated them being perpendicular, like in PCA.
- How is the difference between generation and probe accuracy used in the evaluation?

**Limitations:**

The authors have adequately addressed the limitations and risks of their work.

---

> ### Author Rebuttal · Authors · 2023-08-07
>
> We thank the reviewer for their positive and helpful feedback.
>
>
> ***Addressing questions***
>
>
> **Only 40 samples for probing?**
>
>
> Questions in TruthfulQA have an average of $7.2$ answers. Therefore, the $40$ samples provide roughly $288$ QA pairs to train and evaluate the probe. As shown in Figure 6(A), the performance improvement brought by using more QA pairs plateaus early.
> Figuring out the direction of different subcategories, how they differ, and how they could compose are interesting research questions! We leave it to future research.
>
>
> **Similarities of truthful directions across attention heads**
>
>
> Directions from different attention head output spaces lie in different spaces mapped to each other with complex nonlinear calculations. It's difficult to calculate the similarities between vectors across spaces.
>
>
> **Evaluation on other benchmarks**
>
>
> KL divergence results suggest that the model deviates little from the original language distribution. Pretraining loss (CE) suggests the model is still good for its original purpose, predicting the next-token. The conventional wisdom is that CE is well-correlated with downstream task performance [1, 2]. Nevertheless, what the reviewer raised is an important and solid concern that model-editing method publications should all address. The positive results in our generalization experiment onto TriviaQA and Natural Questions provide initial evidence that our methods don't degrade the model's capability, but extensive experiments like MMLU are more convincing.
>
>
> [1] Liu, Hong, et al. "Same pre-training loss, better downstream: Implicit bias matters for language models." International Conference on Machine Learning. PMLR, 2023.
> [2] http://mitchgordon.me/ml/2020/06/30/pretraining-linearity.html
>
>
> **Using GPT as judge**
>
>
> If I understand correctly, the reviewer is worried by the possibility that the intervention actually takes advantage of a spurious correlation of GPT-judge, so the True*Info scores should not be trusted. Please correct me if I'm wrong! To rule this out, the multi-choice accuracy compares the conditional probabilities of several candidate answers and is free of GPT-judge or such worries. Moreover, we printed out all post-and-pre-intervention QA pairs to be examined in Appendix A. We manually checked a subset of these and found that the truthfulness is improved.
>
>
> **Explain Figure 2(B)**
>
>
> The setup is similar to PCA but relies on a different way to select the base directions. The first truthful direction is normal to the hyperplane that best separates truthful and false activations; the second truthful direction is found with the same process but in a subspace orthogonal to the first truthful direction. By such a setup, it is possible that the major axes of the ellipses are parallel.
>
>
> **Difference between generation and probe accuracy**
>
>
> Before the intervention, the gap between generation truthfulness and the probe on the best attention head is $83.3\%-31.1\%=52.2\%$ and averages to around $40\%$; after the intervention, the largest gap is $83.3\%-49.1\%=34.2\%$, and the average gap is $22\%$.

---

> > ### Comment · Reviewer_Zn8f · 2023-08-13
> > **Reponse**
> >
> > I acknowledge the rebuttal by the authors. I have raised my score from 6->7

---

### Official Review · Reviewer_wYs3 · 2023-07-04

**Soundness:** 4 excellent
**Presentation:** 4 excellent
**Contribution:** 3 good
**Rating:** 8
**Confidence:** 4

**Summary:**

The authors study how to steer the text generation from different LLMs to be more truthful. They do so by finding directions in feature space (for each attention head), that correspond to truthfulness, and intervening at inference time by adding these directions to the activations of the relevant attention heads. They thoroughly characterise the method.

(I gave the score of 7, but I was torn between 7 and 8)

EDIT: Updated to 8 after rebuttals

**Strengths:**

This is a great paper, reading it sparked joy. It's very, very well written; clear and easy to follow. The research question is important and interesting. The experiments are quite extensive, rigorous, and seem to be reported even-handedly (i.e. I don't get the impression that the authors try to display their method in a favourable light, see e.g. the generalisation results.

**Weaknesses:**

Major concerns (I only have one):
- 1. Comparison to related work and novelty: Lots of relevant work is cited, and, AFAICT, the main relevant papers are mentioned. But the authors should do a much better job at explaining how exactly their method is different from key related work, like Subramani, Hernandez, and Burns. In the author's defense, some of the relevant work was published shortly before the NeurIPS deadline and would be considered concurrent. (I somewhat know the literature and think that there is sufficient novelty in this paper; both from a methods point-of-view, and also from the application of an activation-editing method to the field of truthfulness; BUT, it should be explained better)

Medium-sized concerns:
- 2. Why did you not try to fit one single probe to all activation (maybe with high regularisation to get a sparse result). Or, at least, choose which heads you accept with an algorithm that takes into account feature redundancy? This should probably be an ablation, especially as you claim somewhere that a key difference to previous work is that you only modify some activations. So you should study whether this actually makes a difference.
- 3. I think that for a more comprehensive evaluation of generalisation, it would be good to include a multiple-choice dataset where you don't have to generate the false answers yourself first (e.g. like MMLU)
- 4. I feel like the abstract is a bit too enthusiastic, in that it only cites the most impressive results. I appreciate that that's probably a matter of taste, though.
- 5. Several figures and tables: I think I know what the difference between "True" and "MC acc" is, but it's not very well explained. Also, is True evaluated by GPTJudge or humans?
- 6. In Table, 1, I would be interested in the "linear probing" baseline (which you can, of course, only use or multiple-choice). In the intro, you say "we observe a large 40% difference between probe accuracy and generation accuracy"
- 7. There are known issues with GPT-Judge, e.g. that it rates answers that express more uncertainty (even in unhelpful ways) as more truthful. More human evaluation would be better.


Some small suggestions:
* Figure 1: wait, did scholars in the middle ages not think the world was flat? I didn't know that. Maybe take an example that is easier to understand, like "breaking a mirror brings 7 years bad luck"
* Section 3.1: Q is often shorthand for the query matrix, so I'd recommend using a different variable.
* For Figure 2B: I had looked at this figure before reading the text and interpreted the two directions as the coefficients of the probe with the highest magnitude. Clarify in the caption, or refer to text.
* Section 3.1: You should  more clearly explain that TruthfulQA is a dataset that consists of questions that many humans commonly answer incorrectly.
* You should explain how you sample from Llama (temp, topk, ...)
* you write "Intervening in this direction is equivalent to doing a gradient descent on the head activation to maximize its probability of being predicted as truthful" -> this confused me, as it's only equivalent to doing ONE gradient step.
* Figure 4: It would be nice to also see the baseline (alpha = 0 and/or K = 0).
* Figure 4: I'd explain in the FIgure caption or Figure that the second column measures truthfulness and the third column measures how invasive the intervention is.
* I don't understand the following sentence: "We combine the answers from two hold-out sets for evaluation so no test samples are used in direction finding."
* Section 4.1: You should explain that the metric true*informative only applies to the generation part. This is obvious, once you think about it, but it would have helped me understand the paper faster.
* Table 3: Say that this is on Llama (without IFT) again.
* line 275: text fragment

**Questions:**

none

**Limitations:**

yes, limitations are highlighted well

---

> ### Author Rebuttal · Authors · 2023-08-07
>
> We thank the reviewer for their positive and helpful feedback. We will edit accordingly to perfect the reading experience of the paper.
>
>
> ***Addressing major concerns***
>
>
> **More detailed related work**
>
>
> CCS (Burns et al. 2022) is already introduced in Introduction, Related Work as well as the experiment section. But we will add a more detailed introduction to REMEDI and Subramani et al. in the Related Work.
>
>
> REMEDI (Hernandez et al.) learns a parameterized function to intervene the representation space of the language model to edit the knowledge it has, e.g., "Anita is an attorney".
>
>
> Subramani et al. discover that, by manipulating the representation space of GPT-2, we can force it to generate a specific sentence. And the average of such "steering vectors" can achieve sentiment transfer on the Yelp Sentiment dataset.
>
>
> ***Medium-sized concerns***
>
>
> **A single sparse probe**
>
>
> We didn't train a huge probe because we think of selecting attention heads as a natural way of sparsifying intervention. But a probe with strong regularization could serve as an interesting baseline.
> To compare with intervening everywhere, we controlled the number of heads intervened (K) in Figure 4. If we extrapolate the trend to $K=1024$ (all heads), it will be an extremely strong intervention, where the low helpfulness might be undesired.
>
> **Linear probe baseline**
>
>
> In line 153, we mentioned the probe validation set accuracy is $83.3\%$ for linear probes with a 4:1 split on multi-choice QA pairs. It's a good idea to mention the highest probe accuracy trained with $5\%$ of TruthfulQA in Table 1.
>
>
> **Human evaluation**
>
>
> Human evaluation is the ultimate evaluation of truthfulness and helpfulness, but it's also noisy and suffers from a lack of reproducibility. Different authors tried to annotate generated answers, but we found people disagree on what "truthfulness" and "helpfulness" mean for each question. Therefore, we kept only reproducible scores and printed out all QA pairs in Appendix A.

---

> > ### Comment · Reviewer_wYs3 · 2023-08-11
> > **Response**
> >
> > Thanks. I think this response mostly fails to address my concerns, so I won't raise my score (but hey, it's a 7 already!).
> >
> > Re related work, I already acknowledged that you cite most related work, but you fail to explain how exactly your work is different. This is still the case in your response here.
> >
> > Re a single probe: intervening on many heads (each individually fitted) is not the same as fitting one probe to all heads simultaneously; or using any other method that accounts for correlation between the features.
> >
> > And I still think another MC dataset would help the paper.

---

> > > ### Author Response · Authors · 2023-08-11
> > > **Response**
> > >
> > > Re related work, here I spell out the difference in detail. (1) To CCS, the biggest delta is the intervention proposed, whereas CCS focuses on understanding the direction of truth in hidden spaces; (2) To REMEDI, the goal is different, we hope to improve overall truthfulness while REMEDI aims at editing specific knowledge like "Anita is an attorney"; (3) To Subramani et al., the task is different, they focuses on forcing generating one specific sentence and style transfer, while we focuses on controlled open-ended generation.
> > >
> > > Re large linear probe, I guess what you meant by correlation is that features from different heads and layers will compete against each other via a strong regularization term. If we force the 0-norm of the big probe weight to be $128*48=6144$, it will change exactly the same number of activations as in ITI but at different locations. That's indeed a good alternative method to try.
> > >
> > > Hope this answers your questions. We appreciate your insightful review! We will be available during the discussion period if any further questions arise.

---

> > > > ### Comment · Reviewer_wYs3 · 2023-08-12
> > > > **Response**
> > > >
> > > > Thanks. I would want to see experiments on my point 3) (and ideally also point 2) to raise my score.

---

> > > > > ### Author Response · Authors · 2023-08-15
> > > > > **Response to reviewer**
> > > > >
> > > > > Hello, could you have a look at a latest response to all reviewers? Do you think it addresses your third point "And I still think another MC dataset would help the paper"?
> > > > >
> > > > > I'm looking into alternative methods to select intervention points and will keep you posted.

---

> > > > > > ### Comment · Reviewer_wYs3 · 2023-08-15
> > > > > >
> > > > > > Yes, that's a great result (assuming that it wasn't cherry-picked). I will update my score now

---

> > > > > > > ### Author Response · Authors · 2023-08-22
> > > > > > > **Alternative methods to select intervention points**
> > > > > > >
> > > > > > > Hello, we investigated alternative methods to select intervention points and will add the following paragraph into the appendix.
> > > > > > >
> > > > > > > > Here we test two alternative methods for selecting intervention positions. To start with, we concatenate outputs from all self-attention heads across layers and train a single probe to classify truthfulness on them. The resultant accuracy is slightly higher than that from the best single attention head ($84.4\%$ compared to $83.3\%$) and is insensitive to normalization methods, including feature normalization and PCA. In the first alternative method, we intervene on all attention heads, denoted "without selection." In the second one, we select the intervention position by ranking the absolute value of probe coefficients, denoted "point-wise selection." We choose the same amount of features to intervene as in our "head-wise selection" baseline, $K$ times attention head dimensionality. Results are shown below. The conflict between truthfulness and helpfulness can also be found for the two alternative methods, and the optimal $\alpha$ cannot achieve as good performance as baseline methods, demonstrating the importance of sparsification methods in ITI. It also suggests that head-wise selection might serve as a good heuristic for such sparsification.
> > > > > > > |                                          | $\alpha$    | True*Info (\%) | True  (\%)     | MC acc. (\%) | CE    | KL    |
> > > > > > > |------------------------------------------|-------------|----------------|----------------|--------------|-------|-------|
> > > > > > > | Without selection                        | 5           | 35.4           | 37.1           | 28.3         | 2.19  | 0.08  |
> > > > > > > | Point-wise selection                     | 15          | 39.2           | **55.3**       | 28.7         | 4.01  | 1.95  |
> > > > > > > | Head-wise selection                      | 20          | **42.3**       | 45.1           | **28.8**     | 2.41  | 0.27  |
> > > > > > >
> > > > > > > > **Caption**: Comparison with different intervention position selection methods and their respective optimal $\alpha$'s on LLaMA-7B. Results are from 2-fold cross-validation, a different protocol from Table 1.

---

### Official Review · Reviewer_gjfu · 2023-07-06

**Soundness:** 3 good
**Presentation:** 3 good
**Contribution:** 3 good
**Rating:** 7
**Confidence:** 4

**Summary:**

This paper proposed Inference time intervention, which can be used to enhance the trustfulness of LLMs. The method first uses supervised learning to identify latent vectors for factual outputs and then shift activations based on these vectors. They repeat the same intervention aggressively. The proposed method is computationally inexpensive. The experiments on TruthfulQA benchmark demonstrate ITI can boost model performance to a large extent. The authors also verify the generalizability of ITI on other benchmarks and show better performance than baseline methods.


**Strengths:**

The paper proposes a way to improve the truthfulness of LLMs and the method shown to be effective and data efficient. The analysis in the paper is comprehensive and supports the claim: starting from understanding whether a network knows the true answer to proposing ITI to improve the truthfulness. Each step is well explained and transferred smoothly.


**Weaknesses:**

NA

**Questions:**

- How does ITI compress to weight editing since both aim for minimal invasion? Even though in the related work, it said the latter one could reduce general robustness but not sure if that will happen in this application.

- In Fig.2B, it showed there could be more than 1 truthful direction. In ITI, do you consider shifting along different directions or do you only do the first main direction? Will adding more directions help the models more?

---

> ### Author Rebuttal · Authors · 2023-08-07
>
> We thank the reviewer for their positive and helpful feedback. For the raised two questions:
>
>
> **Comparison between ITI and weight editing**
>
> ITI is an activation editing method that does not change model weights. It enjoys the benefit that the intervention strength could be tuned by a hyperparameter $\alpha$, which gives practitioners more flexibility in deploying this method.
> Qualitatively, Appendix A shows that the post-intervention remains language fluency; quantitatively, the shift from pre-intervention language distribution is measured by the KL divergence (KL) and pretraining loss (CE).
> In fact, ITI with a selected $\alpha$ is equivalent to doing sparse editing to the bias term of the output projection MLP. I successfully edited a LLaMA model offline within the HuggingFace framework, which I can release after de-anonymization. I talked with it, and it still speaks fluent, sense-making English. We will welcome everyone to chat with it.
>
>
> **Alternative Truthful Direction**
>
>
> We tried alternative directions like the direction pointing from the mean of false activations to the mean of truthful activations (mass mean shift). Table 3 shows that the “mass mean shift” outperforms the first probe direction.

---

> > ### Comment · Reviewer_gjfu · 2023-08-14
> >
> > I have read the authors' rebuttal and keep my original score.

---

### Official Review · Reviewer_4t4s · 2023-07-07

**Soundness:** 2 fair
**Presentation:** 2 fair
**Contribution:** 2 fair
**Rating:** 6
**Confidence:** 4

**Summary:**

This paper studies the truthfulness for LLMs through the internal representations of models, which is an important and challenging research direction. As previous works have demonstrated that LLMs can contain truthful information internally despite giving an incorrect output, this paper proposes the Inference-Time Intervention (ITI) technique that shifts the activations of several selected attention heads when generating tokens. The authors run experiments mainly on the TruthfulQA dataset to demonstrate the effectiveness of ITI, and analyze the generalization ability on two other datasets. Comparisons with other techniques, including few-shot prompting, finetuning, and CCS, have been studied.

**Strengths:**

Overall, this paper is studying an important problem with contemporary LLMs. Several strengths I particularly admire are:

- The methodology of this paper is appealing. Rather than merely finding an internal representation like CCS, this paper goes further and keeps the token generation process unchanged. This allows the ITI technique to be integrated with existing standard pipelines, and is less invasive.
- The intuition of ITI is straightforward. In fact, shifting the activations of attention output linearly is at least intuitive, and the fact that computation cost will not be influenced is quite preferred.

**Weaknesses:**

Notwithstanding its strengths, this paper presents several weaknesses that prevent its current acceptance:

- The first contribution of the paper, the exploration of misalignment between the outputs and the internal information, has been previously addressed in studies such as the CCS paper. The analysis in Section 3, while valuable, does not offer substantial novel insights. These points should not be viewed as the principal contributions of the paper and should be rephrased accordingly.

- A major limitation of the paper is its narrow focus on a limited number of datasets, which doesn't convincingly support the effectiveness of ITI. Most experiments are conducted on the TruthfulQA dataset, with a minimal generalization to two additional datasets. This leaves it uncertain whether ITI is universally applicable or specific to TruthfulQA. More tests on a variety of datasets would strengthen the paper.

- When focusing solely on the TruthfulQA dataset, ITI's improvement appears marginal. For instance, the True*Info accuracy only improves by 2% with few-show prompting, while few-show prompting itself provides a more significant 13% improvement. These findings suggest that the benefits offered by ITI could largely be achieved through few-show prompting alone. Therefore, the paper's claim that ITI is an important technique to consider needs more compelling support.

- Finally, the method for determining the hyper-parameter $\alpha$ remains ambiguous. The crucial question is whether $\alpha$ is universally applicable or requires case-by-case optimization. Considering the slight improvement provided by ITI, it is clear that selecting an appropriate $\alpha$ is vital, as also suggested by Fig 4. However, the paper fails to explain how this selection should be done.

**Questions:**

The following questions would not change my rating, but I will highly appreciate if the authors can fix them:

- In line 46, the authors mention the prove accuracy of the model. However, it seems to me that this is not a property of the LLMs, but is highly relevant to the classifier you use. I'd recommend the author not to introduce this as a "concept" (since it only appear once in line 187), but just describe it.
- The value of CE and KL is uninformative. From the table, I cannot tell whetehr a CE value is high or not, and I am not sure whether they are important. I recommend the authors to clarify the meaning of their value, and explicitly talk about what we can learn from the variation.
- It seems that the way this paper use CCSis inconsistent with the original CCSpaper. First, I don't think using only *one* sample is enough to find a good direction. Second, it is not designed to shift the activation of attention heads, but rather the output of each *layer*. I am not sure whether it is appropriate to use CCS in this way.
- Line 275, typo.



**Limitations:**

Not applicable.

---

> ### Author Rebuttal · Authors · 2023-08-07
>
> We thank the reviewer for their useful and constructive feedback. We have clarified several elements of the paper below and will incorporate them in our updated version.
>
>
> ***Addressing Weakness***
>
>
> **Figure 2 not a contribution**
>
>
> Though (orthogonal) probe techniques have been discussed in the literature, applying it to LLaMA and TruthfulQA hasn't been done before. Thus we need to introduce this procedure to lay the ground for introducing ITI. We will edit the prose and be precise in claiming contributions.
>
>
> **Only on TruthfulQA dataset**
>
>
> Thanks for bringing up this concern. Testing in additional settings would certainly strengthen the result. However, it's worth noting that the experiments in the paper do cover a variety of contexts, for two reasons:
> + TruthfulQA is a diversified dataset covering 38 subcategories. They range from law, stereotypes, to conspiracy theories, etc. We did not treat subcategories differently but, as Figure 5 shows, ITI improves most of them. This is nontrivial in itself.
> + The zero-shot generalization experiment on TriviaQA and Natural Questions is the most rigorous test for ITI since we apply the intervention learned on TruthfulQA on new datasets without any modification. It's a true zero-shot setting and even a small improvement in such a setting is positive.
>
>
> **Marginal improvements**
>
>
> We hope to propose a technique that could be added onto real ChatBot products like ChatGPT or Claude. Few-shot prompting takes up precious context length and significantly shifts original language distribution. Our focus in Table 1 through Table 3 is the delta between methods with and without ITI, e.g., from $30.5\%$ to $42.3\%$ for LLaMA baseline.
>
>
> **Selecting $\alpha$**
>
>
> In the purely research-oriented case of this paper, we do train, validation, and test split and choose $\alpha$ to optimize the True*Info metric on the validation set. Genealization experiments on TriviaQA and Natural Questions show that a good $\alpha$ tuned on TruthfulQA has the potential to work as well on other domains. That is to say: $\alpha$ is more dependent on the specific model rather than requiring careful tuning for specific datasets.
> However, the real-life dilemma is that we are unsure what practitioners are optimizing for. The $\alpha$ should be selected per need by the user via trial and error: if users are extremely cautious about untruthful replies, $\alpha$ should be tuned up; otherwise, if helpfulness is also a requirement. We will clarify this point in the paper.
>
>
> ***Addressing Questions***
>
>
> **Writing around line 46**
>
>
> This is a valid point. We will change it into "We look into this through the difference between..."
>
>
> **How to interpret CE and KL?**
>
>
> We will better contextualize the value, e.g., equivalent to losing how many parameters or how much less training data.
>
>
> **CCS usage**
>
>
> In Table 3, all directions, e.g., CCS, and mass mean shift, are found using half of TruthfulQA. What do you mean by "one sample"?
> Thanks for pointing out the confusion with the CCS baseline. The issue here is that ITI focuses on attention heads, while CCS focuses on MLP activations. As a result, we compare with a CCS analog for attention heads. We will clarify this.

---

> > ### Comment · Reviewer_4t4s · 2023-08-17
> >
> > Thanks for the response. I appreciate the supplementary experiment. It addresses my major concern about whether ITI can be trained in advance and generalize to various settings. I will raise my score to 6.
> >
> > > "We will edit the prose and be precise in claiming contributions."
> >
> > Can you please explicitly let me know what you will claim?
> >
> > ---
> >
> > I believe your work will be benefited from the following improvements:
> > - Some theoretical explanation or at least some intuition on why ITI can improve truthfulness. The reason why I previously requested more experiments is that we do not know the *boundary* of ITI, i.e., under what setting its effectiveness vanishes. To systematically address this, some theoretical analysis or ablation study can be helpful for understanding the value and limitations of ITI.
> > - Better analysis of the incorporation with existing techniques. It has been shown in TruthfulQA that when equipped with FSL, the improvement of ITI becomes marginal (but still positive, I agree). Is this true in MMLU / TriviaQA as well? More experiments and analysis on this will be insightful.

---

> > ### Author Response · Authors · 2023-08-17
> > **Response from authors**
> >
> >
> > Thank you for acknowledging the new generalization results.
> >
> > **Phrasing Contribution**
> >
> > I changed the paragraph in line 63 into:
> > > This work makes two main contributions. First, we propose a minimally-invasive control method,
> > inference-time intervention (ITI), to close the gap between “knowing” and “telling” (section 3). ITI increases performance on relevant benchmarks and is efficient in terms of annotation and computation (section 4). Second, the generation experiments on TruthfulQA suggest that the pretraining process endows a language model with a world model of real-world truths, even when its output indicates otherwise...
> >
> > Note that the second contribution is not about Figure 2 but rather a scientific understanding drawn from the results of aforementioned engineering techniques (first contribution).
> >
> > I also changed the paragraph in line 135 into:
> > > Following works that finds interpretable directions within activation spaces of neural networks, we investigate whether there are vectors in the activation space of transformer layers that correspond to ``truthfulness'' by applying existing techniques: probing and orthogonal probing.
> >
> > **New Suggestions**
> >
> > We appreciatively agree that these are important improvements that could be made. I could guess the _boundary_ of ITI is the accuracy of the internal world model endowed by pretraining, which is unknown and could only be approached via ever-advancing inference-time techniques. I will also look into the relationship and comparison of ITI and existing techniques to better establish it as a mature ML technique.

---

### Author Response · Authors · 2023-08-15
**MMLU Evaluation**

We thank all reviewers for their insightful comments. Following suggestion by reviewer 4t4s and wYs3, we did more evalution expriment and the results will be updated in future version of the paper.

|                   |  MMLU | TruthfulQA |
|-------------------|:-----:|:----------:|
| LLaMA1-7B w/o ITI | 35.71 |    34.33   |
| LLaMA1-7B w/ ITI  | 40.16 |    61.50   |

In the table above, we compare the original LLaMA1-7B with and without our ITI methods. The ITI is trained solely on TruthfulQA and no (hyper)parameters or prompts are tuned for MMLU. We notice that ITI not only boost truthfulness on TruthfulQA but also zero-shotly generalize to MMLU, an aggregated benchmark that covers 57 subjects across STEM, the humanities, the social sciences, and more.

The baseline results are taken from this [leaderboard](https://huggingface.co/spaces/HuggingFaceH4/open_llm_leaderboard); the ITI results are produced using the same code that produced this leaderboard. Note that because the leaderboard doesn't support customized model, we cannot upload our score. This is also in line with NeurIPS's anonymization policy. Also note that the TruthfulQA score presented in this leaderboard is `MC2`, while the multi-choice score in our paper is `MC1`; `MC1` presents only one truthful answer in its choices while `MC2` presents multiple truthful answers; these are two different scores and not comparable to each other.

We will be available during the discussion period if any further questions arise.

---

### Decision · Program_Chairs · 2023-09-21

**Decision:**

Accept (spotlight)

**Comment:**

This paper proposes a technique named Inference-Time Intervention (ITI) to enhance the truthfulness of large language models (LLMs). ITI shifts model activations during inference, following a learned set of directions across a limited number of attention heads. The reviewers applauded the problem under study is well-motivated, the technique is less invasive and computationally inexpensive, and the writing is clear and easy-to-follow. They shared concerns on mainly using TruthfulQA for experiments, difference from related work and novelty, and also raised questions about experimental details and improvement brought by the proposed techniques. In general, the reviewers are satisfied with the authors' responses where they added more experiments and clarified other questions. I would recommend the paper to be accepted as spotlight, provided that it will be revised as discussed in the rebuttal phase.